# Biomimetic Remineralized Three-Dimensional Collagen Bone Matrices with an Enhanced Osteostimulating Effect

**DOI:** 10.3390/biomimetics8010091

**Published:** 2023-02-23

**Authors:** Irina S. Fadeeva, Anastasia Yu. Teterina, Vladislav V. Minaychev, Anatoliy S. Senotov, Igor V. Smirnov, Roman S. Fadeev, Polina V. Smirnova, Vladislav O. Menukhov, Yana V. Lomovskaya, Vladimir S. Akatov, Sergey M. Barinov, Vladimir S. Komlev

**Affiliations:** 1Institute of Theoretical and Experimental Biophysics, Russian Academy of Sciences, Pushchino 142290, Russia; 2Baikov Institute of Metallurgy and Materials Science, Russian Academy of Sciences, Leninskiy Prospect 49, Moscow 117334, Russia

**Keywords:** bioactive materials, bone grafts, scaffolds, calcium phosphates, coatings, bone tissue engineering, remineralization, demineralized bone matrix

## Abstract

Bone grafts with a high potential for osseointegration, capable of providing a complete and effective regeneration of bone tissue, remain an urgent and unresolved issue. The presented work proposes an approach to develop composite biomimetic bone material for reconstructive surgery by deposition (remineralization) on the surface of high-purity, demineralized bone collagen matrix calcium phosphate layers. Histological and elemental analysis have shown reproduction of the bone tissue matrix architectonics, and a high-purity degree of the obtained collagen scaffolds; the cell culture and confocal microscopy have demonstrated a high biocompatibility of the materials obtained. Adsorption spectroscopy, scanning electron microscopy, microcomputed tomography (microCT) and infrared spectroscopy, and X-ray diffraction have proven the efficiency of the deposition of calcium phosphates on the surface of bone collagen scaffolds. Cell culture and confocal microscopy methods have shown high biocompatibility of both demineralized and remineralized bone matrices. In the model of heterotopic implantation in rats, at the term of seven weeks, an intensive intratrabecular infiltration of calcium phosphate precipitates, and a pronounced synthetic activity of osteoblast remodeling and rebuilding implanted materials, were revealed in remineralized bone collagen matrices in contrast to demineralized ones. Thus, remineralization of highly purified demineralized bone matrices significantly enhanced their osteostimulating ability. The data obtained are of interest for the creation of new highly effective osteoplastic materials for bone tissue regeneration and augmentation.

## 1. Introduction

Reconstructive surgery of bone defects resulting from trauma, complex dental extirpation, resection of osteomyelitis or malignant neoplasms (osteosarcomas, metastases), etc., is the most actively developing area of modern regenerative medicine [1,2,3,4]. To create optimal conditions for bone tissue regeneration, there is a great need for porous, biocompatible and stimulating localized osteogenesis materials that can provide effective recovery of bone tissue structure and volume with complete regeneration of bone integrity as an organ, and preferably within a short time frame [5,6]. Currently, in traumatology, orthopedics, and dentistry, using the technology of reconstructive surgery, the most popular materials are autografts, allografts and bioartificial synthetic calcium phosphate (CP) materials [7,8,9]. 

Spongy autografts are still the “gold standard” for guaranteed bone tissue regeneration; however, two natural problems arise when using them: a hard limit on the amount of bone tissue possible for sampling and additional traumatic loads on the patient’s body [10,11,12,13]. Allografts from donors or cadavers from bone banks and their demineralized bone matrix (DBM), or partially demineralized bone matrix (pDBM), derivatives serve as alternatives to autografts in the reconstruction of bone [14,15]. In turn, the use of allografts is significantly limited by the risk of infection transmission and the development of rejection in the recipient’s body [16,17,18]. At the same time, although the use of pDBM provides a more significant clinical effect compared to DBM, its use is also aggravated by the risk of infection transmission due to the inability to remove pathogens from the mineral part of the bone tissue [12]. DBM (due to decellularization) is characterized by a minimal risk of MHC-mediated immunogenicity of pathogen transmission, but its use is limited by weak osteoinductivity and unstable osteostimulatory effects due to the rapid resorption in the recipient’s body [4,11]. In the case of the latter, it should be noted that the rapid degradation of DBM is caused primarily by damage to the ultrastructure and microarchitectonics of the extracellular matrix, leading to the fact that from a full-fledged bioactive osteoconductor/osteoinducer, DBM turns into an alarmin or even “danger signal”, triggering a DAMP-immune response with a subsequent rejection in the recipient body [19]. It is important to understand that in normal conditions, the bone matrix is not just an inert three-dimensional scaffold, but is a complex, spatial, self-regulating structure, performing both support and signaling functions, and possessing haptotaxis that stimulates the attraction and differentiation of progenitor cells into the osteoblast phenotype under the action of specific proteins and functional peptides of the ECM [20]. 

Thus, for example, the most important collagen-binding proteins, integrins, and other important collagen receptors include the discoidin domain receptor (DDR1 and DDR2), platelet glycoprotein GPVI, immune receptors, the plasma protein von Willebrand factor (vWF) and other macromolecules and proteins of the ECM, provide chemotaxis and are the mediators of cell adhesion, cell migration, and cell differentiation progenitor cells [21,22,23,24,25,26]. In addition to the above, acellular bone extra-cellular matrix, an essential organic constituent of bone tissue, usually consists of 100 different structural and signal proteins, of which type I collagen is the main one, and small amounts of glyco-proteins, proteoglycans, and noncollagenous proteins including osteocalcin (OCN), osteopontin (OPN), osteonectin (ON), bone sialoprotein (BSP), fibronectin (FN), vitronectin (VTN), bone acidic glycoprotein-75 (BAG-75), matrix Gla-protein (MGP), thrombospondins, etc., as well as important growth factors, such as bone morphogenetic protein growth factors/transforming growth factor beta (BMPs/TGFβ) superfamily, fibroblast growth factors (FGFs), vascular endothelial growth factor (VEGFs), and fibroblast growth factor 23 (FGF23) [27,28,29]. Alongside that, specific lipids of the ECM, such as some fatty acids, cholesterol, phospholipids, oxysterols, etc., have also been purported to act on bone cell survival and functions, the bone mineralization process, and critical signaling pathways [30]. 

In addition to the above, it is due to the specific structure of the ECM collagen fibrils that natural biomineralization of bone tissue occurs, in which specific conformations of “bare” zones of type I collagen provide passive precipitation of amorphous calcium phosphates with their further transformation into hydroxyapatite (HAp) [31], that together provide unique rates of bone strength and elasticity.

In turn, the active use of synthetic calcium phosphate compounds is associated with their similarity to the mineral component of bone tissue, and has been shown in a number of works regarding osteoconductive and osteoinductive action [32,33,34]. The key factors determining the osteogenic potential of such materials are the method of their synthesis and physicochemical characteristics, such as phase composition, particle size, and shape, solubility, etc. Numerous traditional methods exist today, for example, sintering of pellets, foaming, injection, and subsequent dissolution of special pore-forming additives, etc., to form various porous structures of calcium phosphate ceramics [8]. The main disadvantage of these methods is their long time and energy consumption, as well as the low reproducibility of spatial and dimensional parameters (especially the internal micro- and macrostructure) of created products. Ceramics obtained by high-temperature solid-phase synthesis do not have the necessary osteogenic properties and are not resorbed in the body, so the body’s response to them is limited to either the formation of a fibrous capsule around the material, or the development of an inflammatory response to calcium phosphate particles formed during implant wear, which can also lead to their rejection [35,36,37]. Recently, nanosized HAp, as well as HAp precursors—octacalcium phosphate (OCP) and dicalcium phosphate dihydrate (DCPD)—were considered the most promising among all calcium phosphate compounds due to their similarity to the mineral component of bone tissue and proven osteoinductive properties [38,39,40].

Thus, the effective osteoplastic material can be low-temperature HAp precursors of highly purified nonimmunogenic extracellular bone matrix, with preserved fibrillar structure and spatial architectonics, composite biomimetic material, which regulates directly the processes of cell migration, proliferation, differentiation, biomineralization and bone remodeling/regeneration, directly in the place of implantation, due to its natural composition and structure [41]. Calcium phosphate production through a low-temperature approach mimics the natural CPC of native bone tissue at its best. Precursors of Hap, DCPD and OCP, synthesized under conditions that maximally imitate the natural (physiological) process of biomineralization, can have a direct inducing effect similar to native CPC. It is known that in conditions of deposition from the internal environment of the organism, these precursors act as centers of crystallization of bone hydroxyapatite and dental enamel, resulting in the formation of precipitated amorphous Hap, and subsequently HAp with a high degree of crystallinity. Interest in these materials is also related to the possibility of the introduction of biological agents and directed chemical functionalization in the process of their synthesis, with further controlled release of the necessary elements to activate and maintain regenerative processes of the body. The proposed technology of material creation consists of the low-temperature method of the precursor deposition of HAp, DCPD, under conditions maximally close to physiological, by the type of primary mineralization of the bone tissue.

The first attempt to create such a material based on demineralized extracellular bone matrix, with a maximally preserved ultrastructure and the possibility of fine regulation of physicochemical parameters of low-temperature DCPD deposition, was undertaken in this work, and the effectiveness of the proposed approach was evaluated under in vitro and in vivo conditions. 

## 2. Materials and Methods

### 2.1. Preparation of Demineralized Bone Matrices

The initial reagents were purchased from Sigma-Aldrich (Saint Louis, MO, USA) and used as received. All chemicals and solvents, except for those stated below, were purchased commercially and used without further purification, unless otherwise stated.

Demineralized bone matrix (DBM) was obtained according to the author’s method (patent RU 2686309 C1, 04.25.2019). Multistage processing of xenogeneic cancellous bone tissue of mature bovines was carried out, including complete decellularization and demineralization of bone tissue with maximum preservation of the structure and microarchitectonics of the fibrillar collagen matrix. After carrying out all the necessary procedures, decellularized and demineralized three-dimensional porous bone collagen blocks, with axial dimensions of 1 × 1 × 0.5 cm, were obtained. For in vitro and in vivo experiments, the obtained blocks were additionally sterilized by incubation in sterile phosphate-buffered saline (PBS), with the addition of antibiotics and antimycotics: gentamicin sulfate (0.02 mg/mL) (Sigma-Aldrich, Saint Louis, MO, USA), fluconazole (0.04 mg/mL) (Pfizer, Paris, France) and ciprofloxacin (0.008 mg/mL) (Sigma-Aldrich, Saint Louis, MO, USA), at a sample volume to solution volume ratio of 1:20. The blocks placed in PBS with antibiotics were incubated for 48 h with constant stirring and a temperature of 37 °C in a shaker-incubator (Biosan, Riga, Latvia). After that, for 24 h, the blocks were washed three times in sterile PBS, pH value 7.4. 

### 2.2. Quantification of Calcium and DNA Content

For each group of samples before and after decellularization, 10 mg tissue fragments (n = 5) were taken and homogenized in microtubes using a Merck pestle microhomogenizer (MilliporeSigma, Burlington, MA, USA). DNA extraction from tissue homogenate was performed using the DNeasy Blood and Tissue Kit (QIAGEN, Limburg, The Netherlands). An empty test tube was used as a control, with which the same manipulations were performed as with the experimental test tubes. DNA in the obtained solutions was measured on a NanoVue Plus spectrophotometer (Biochrom, Holliston, MA, USA) at 260 nm.

Calcium content in control (native bone) and DBM samples was determined by absorption spectroscopy before and after implantation. The material samples were dried for 12 h at 90 °C in a hot-air sterilizer (Binder, Tuttlingen, Germany), after which the dry weight of the samples was measured. Then each sample was dissolved in 1 mL of 1 M HCl for 24 h at 20–25 °C. The amount of mineralized calcium was measured using a standard calcium determination kit, Calcium AS DiaS Arsenazo III (DiaSys, Holzheim, Germany). Optical density was measured using a microplate reader Infinite F200 (Tecan, Männedorf, Switzerland). Calcium mineralization values in the samples (µg calcium per mg of sample dry weight) were calculated according to the manufacturer’s instructions. 

### 2.3. Remineralization of DBM

There were three groups of samples obtained in this part of the work (Figure 1). 

The first group represented samples of pure DBM without the use of CPC and was the comparison group (control group). 

Samples of the second group were DCPD powders without DBM: obtained in a 0.5 M NH_4_H_2_PO_4_ solution. To the solution was added 0.9 M CaCl_2_. The formation of DCPD particles was carried out for 24 h, without stirring, at 35 ± 2 °C.

The third group was a DBM with a DCPD coating deposited on it, with a variation in the deposition process conditions and the concentration of the sodium acetate solution. This method of mineralization is based on a regulated change of pH during sedimentation followed by crystallization of calcium phosphate phases, with Ca/P ratio of 1.33–1.67, with a complex of specific properties (biocompatibility, controlled release of necessary elements to activate and maintain regenerative processes of the body). A unique approach to the precipitation of calcium phosphate compounds is based on the classical synthesis of CPC, but in a medium with adjustable pH solutions, sodium acetate, glutamic, and orthophosphoric acid (pH environment from 4 to 8, varied by the acid concentration or by adding alkali to the solution), and with the addition of excess calcium ions to the system. Variation of the phase composition of the coating, its spreading, thickness, particle size, and shape can be achieved by changing the deposition conditions. Deposition of the DCPD coating on the matrix was carried out by changing the concentration of the sodium acetate solution to values of 1, 1.5, and 2 M. The formation of CP coatings is realized by the deposition of DCPD from the solution. The main method of DCPD formation is precipitation from a solution of 1 to 2 M sodium acetate, 0.15 M L-glutamic acid with orthophosphoric acid adjusted to pH = 5.5, adding 10 wt.% CaCl_2_ solution for 48 h at 35 ± 2 °C and constant low-intensive stirring in shaker-incubator. At the first stage of mixing solutions, precipitation of CP does not occur within 60–120 min due to the processes of dissociation occurring in the solution. Upon reaching the required concentration of cations and anions in the near-surface layer, crystallization of CP occurs in the form of DCPD crystals over the surface of the sample. At prolonged soaking in the solution, a large number of crystallization centers are formed evenly over the entire surface and in the second stage, active crystal growth begins. The evenness of the precipitation of DCPD on DBM is achieved by a continuous mixing of the solutions, as well as by using a flowing system of constant circulation of the solution through the polymer matrix. Such materials, due to this preparation method, can have increased biocompatibility compared to traditional ceramic materials. Moreover, no additional surface modification is required, which ensures the preservation of the polymer matrix, and its biological and physicochemical properties.

In addition, for cytotoxicity assays, the HAp powder was synthesized by the classical method of liquid-phase precipitation from solutions, followed by high-temperature treatment described in literature data [9].

### 2.4. Morphological Characterization and Elemental Analysis

Morphological specifics of all samples before and after remineralization were studied using a microscope Tescan VEGA III (scanning electron microscopy (SEM), Brno, Czech Republic), equipped with energy dispersive spectroscopy systems (EDS; INCA Energy Oxford Instruments, Abingdon, UK) to analyze the chemical composition, and samples were previously covered with gold by Q150R Quorum Technologies (Lewes, UK). Surface images of the materials were obtained at a pressure of 7.3 × 10^−2^ Pa in the column and 1.5 × 10^−1^ Pa in the chamber. Energy-dispersive X-ray spectroscopy (EDS) (Oxford AZtecO 4.3 software) was used to investigate impurities on the bone matrix surface.

X-ray diffraction (XRD) was performed to evaluate the crystallographic structure of the coating using an X-ray diffractometer «SHIMADZU XRD-600». The diffraction patterns were studied in the 2θ range from 10° to 45° at a tube voltage of 40 kV and a current of 100 mA. The diffraction peaks were determined by comparison with literature data and consideration of the ICDD database (International Centre for Diffraction Data Power Diffraction File; 2 Campus Blvd: Newtown Square, PA, USA, 2007).

The infrared (IR) spectra of DCPD were recorded on an Avatar 330 FT-IR spectrometer by Nicolet (England), in the 4000–400 cm^−1^ wavelength region. A small amount of powder was mixed with potassium bromide (1 mg of powder in 50 mg of spectroscopic-grade KBr) and pressed into a pellet. The pellet was analyzed in the transmission mode in the main box at room temperature. 

The infrared data for brushite are compiled in Table 1 and compared with the spectrum obtained and reported at 300 and 77 K by Petrov et al. [42]. The spectra are essentially identical with the published data for brushite [43,44,45].

### 2.5. MicroCT Analysis

Microcomputer tomography (microCT) was carried out on microtomography Bruker “SKYSCAN 1275” (Germany), with a resolution of 4.5 microns, providing a detailed analysis of the morphological and density characteristics of porosity and thickness of materials in the Comprehensive ΤΕΧ Archive Network (CT-an) program. The images were captured with 0.73° at each pace, 13.76 μm voxel, and further reconstructed using NReconTM v.1.6.8.0, SkyScan, 2011 (Bruker micro-CTTM, Kontich, Belgium). Ring artifact and beam hardening corrections were applied in reconstruction. Afterward, the reconstructed images were realigned using the Data Viewer TM 1.4.4.0 software (Bruker micro-CTTM, Kontich, Belgium). This method was used to study the properties of specimens and their internal architectonics.

### 2.6. Cell Culture

Human osteoblast-like cells, MG-63, were obtained from the ATCC (Manassas, VA, USA). The cells were cultivated in the EMEM nutrient medium (Sigma-Aldrich, Milwaukee, WI, USA), supplemented with heat-inactivated fetal bovine serum (Gibco, Waltham, MA, USA) to a final concentration of 10% and 40 μg/mL gentamicin sulfate (Sigma-Aldrich, St. Louis, MO, USA), under conditions of 5% CO_2_ content in the air and at 37 °C. The cell cultures were tested for mycoplasma infection using the MycoFluor™Mycoplasma Detection Kit, and no mycoplasma was detected.

### 2.7. Cytotoxicity Assays

In vitro experiments with human osteoblast-like MG-63cells were seeded in an amount of 5 × 10^3^ cells in 100 µL of complete growth medium into 96-well plates at different concentration s of calcium phosphate compounds (Corning Inc., Corning, NY, USA). After 24 h of cultivation, the medium was re-placed with 100 μL of medium containing the above-mentioned CPC, at concentrations of 10, 3, 1, 0.3, and 0.1 mg/mL, and the cultivation was continued for 24 and 72 h. The cells in the control conditions were cultured in the medium without the addition of CPC. Then, cell viability was analyzed, as well as morphological analysis of the cell culture conditions.

The cytotoxicity of CPC and the morphological state of the cell culture were analyzed by in vivo staining of cells with the fluorescent dyes Hoechst 33342 (stains blue nuclei of living and dead cells), propidium iodide (stains red nuclei of dead cells), and calcein AM (stains green cytoplasm of living cells). Cells were stained by adding 1 μg/mL Hoechst 33342 (Sigma-Aldrich, St. Louis, MO, USA), 1 μg/mL propidium iodide (Sigma-Aldrich, St. Louis, MO, USA), and 2 mM Calcein AM (Sigma-Aldrich, St. Louis, MO, USA) to the culture medium. The staining was performed in a CO_2_ incubator for 30 min at 37 °C and 5% CO_2_ in the air.

Microscopic analysis of the stained cell cultures and micro images was made on a Nikon Eclipse Ti-E (Nikon, Tokyo, Japan). The plate with cells and examined samples was transferred to a microscope chamber at 37 °C and 5% CO_2_ content. Cytotoxicity was analyzed by calculating the number of live and dead cells per field of view using the ImageJ software (https://imagej.nih.gov/ij/ (accessed on 19 April 2022)).

### 2.8. Confocal Microscopy

Cytotoxic and osteoconductive properties of three-dimensional DBM, after cultivation with MG-63 cells, were studied by confocal microscopy. Cells cultured on a coverslip were used as a control. DBM and glasses were placed in a Petri dish, a culture medium was added, and cells were seeded on the glass surface at a density of 1 × 10^4^ cells/cm^2^.

Cytotoxic test and morphological analysis of cell adhesion and spreading on the surface of the three-dimensional matrices were performed after 24 and 72 h following the moment of cell seeding on the material, using lifetime staining of cells with fluorescent dyes: propidium iodide (stains dead cells) and calcein AM (stains the cytoplasm of living cells). Cells were stained by adding 1 μg/mL propidium iodide (Sigma-Aldrich, St. Louis, MO, USA) and 2 mM calcein AM (Sigma-Aldrich, St. Louis, MO, USA) to the culture medium. The staining was performed for 20 min at 37 °C. The DBM and coverslips were transferred to a pure culture medium after staining, and cytotoxic test and morphological analysis of cell adhesion and proliferation were performed using a TCS SP5 confocal microscope (Leica, Wetzlar, Germany). Digital processing of confocal photographs was provided using the ImageJ software (https://imagej.nih.gov/ij/ (accessed on 19 April 2022)).

### 2.9. Animals and Surgical Procedures

Eighteen Wistar male rats, weighing 200–220 g (age two months), were used. Animals were individually housed in a temperature-controlled room (22 °C) and fed a standard diet, with full access to water and food. The experiments were carried out according to the Regulations for Studies with Experimental Animals (Decree of the Russian Ministry of Health of 12 August 1997, No. 755). The protocol was approved by the Commission on Biological Safety and Ethics at the Institute of Theoretical and Experimental Biophysics, Russian Academy of Sciences (March 2022, protocol N26/2022). For the experiments, rats were divided into three groups (six in each group) and independent replicates were done for each group. 

The model of ectopic (subcutaneous) implantation of biomaterials was used to study the osteogenic potential of the obtained matrices in vivo. In case of the need to confirm the osteoinductive and osteogenic potential of materials, this model best reflects the effects sought, as it provides results initiated by the material itself, rather than the influence of the native bone microenvironment. The definitions of osteoinduction were thoroughly reviewed by Barradas and co-authors in 2011 [46]. It is generally accepted that an osteoinductive material should induce bone formation upon implantation in non-osseous sites, also known as heterotopic or ectopic sites [46,47,48,49,50]. Thus, confirmation of in vivo bone formation in ectopic sites is necessary before a material is classified as osteoinductive, and even more so as osteogenic. Therefore, a heterotopic implantation model (under the skin, into the muscle, etc., but not into the bone) is the most reliable for revealing the osteoinductive and osteogenic potential of the materials under development. 

The surgeries were performed under general anesthesia with Xylazine 13 mg/kg (Interchemie, The Netherlands) and Zoletil 7 mg/kg (Virbac, Carros, France). To implant the specimens, a 1.5 cm wide transverse skin incision was made in the dorsal interscapular area, and subcutaneous pockets were formed parallel to the skin using a smooth trocar, followed by implantation of the specimens at a depth of at least 2 cm from the incision line. Samples were implanted with full interstitial contact without restriction chambers or meshes. The first group of rats was implanted with DCM. The second group was implanted with DCPD, the third group was implanted with biomimetic DBM + DCPD matrices. The size of the implanted samples was 1 × 1 × 0.5 cm. For post-surgical recovery, the animals were exposed to a heating plate until awakening.

The animals from each group were randomly divided to be euthanized (carbon dioxide protocol) after 7 weeks (50 days) of the surgical procedure. Immediately after humane euthanasia, to prevent autolysis instantly after the withdrawal, samples of implanted materials, with surrounding tissues of the recipient bed, were washed for 30 s with a cold (14 °C) isotonic solution, and fixed for 48 h in neutral buffered formalin (NBF) at the tissue-volume fixator volume ratio 1:30.

### 2.10. Histological Analysis

After the termination of fixation, the fragments of samples were washed with distilled water (3 × 3 min) to remove excessive phosphates and placed for no less than 12 h in medium optimum cutting temperature (O.C.T.) compound Tissue-Tek (Sakura, Tokyo, Japan). Cross slices of the heart ventricles of rats (9 μm) were prepared by cryo-sectioning (MEV SLEE medical GmbH, Germany). The staining of the samples was performed by a conventional method using H&E (Mayer’s Hematoxylin and Eosin Y), and differential staining for calcium deposits alizarin red S (by the McGee-Russell method [51]) and collagen/non-collagen structures (by Lillies trichrome method) [30]. The micrographs of the stained histological samples were obtained on a Nikon Eclipse Ti-E microscope station (Nikon, Tokyo, Japan) and processed using the software NIS Elements AR4.13.05 (Build 933).

### 2.11. Statistical Analysis

Results are presented as the mean ± standard deviation (M ± SD). Each of the in vitro experiments was carried out at least five times (n ≥ 5). The statistical significance of the difference was determined using one-way ANOVA, followed by multiple Holm–Sidak comparisons, *p* < 0.05. The design of the experiment and related statistics (ANOVA) were carried out using SigmaPlot™ 14.0 (Systat Software Inc., San Jose, CA, USA). Plots were created using SigmaPlot™ 14.0.

## 3. Results

### 3.1. Results of Efficiency Evaluation of Bone Matrix Demineralization

#### 3.1.1. Purity Assessment of Demineralized Bone Matrices

After multistep treatment of xenogenic bone matrix (decellularization and demineralization of bone tissue protocol), it was found that residual calcium content in the samples did not exceed 0.78 ± 0.34 µg/mg of tissue dry mass (Figure 2a), with a residual donor cell DNA content of 2.18 ± 0.97 ng/mg tissue mass (Figure 2b), which indicated a high degree of purification of the xenogenic collagen bone matrix. Elemental analysis of the DBM surface revealed the complete absence of impurities on the surface of the samples and confirmed the maximum efficiency of the demineralization performed on the samples (Figure 2c).

#### 3.1.2. Assessment of Preservation of Structure and Architectonics of Three-Dimensional DBM

MicroCT (Figure 3a) and SEM (Figure 3b,c) data of DBM samples revealed that the fibrillar structure of the trabecular matrix of DBM is fully preserved, with no signs of delamination and separation of fibrils or disruption of lamellae integrity with loss of three-dimensional structure were detected (Figure 3b,c), i.e., architectonics of the DBM extracellular collagen matrix completely corresponded to the native collagen extracellular matrix.

Differential histochemical analysis of DBM showed a complete absence of donor cells/residual cell debris in the matrix (Figure 3d), as well as the absence of any calcium compounds in the matrix, including inside the bone trabeculae (Figure 3f). The analysis also confirmed the preservation of the macromolecular collagen structure of trabecula ossea throughout the samples, without any signs of changes in the collagen staining affinity (Figure 3e).

### 3.2. CP Evaluation Results

#### Comparative Analysis of CP

According to the results of XRD, the solution mixing leads to the formation of a single-phase product corresponding to DCPD. The results of scanning electron microscopy showed that DCPD crystals, mostly have no obvious shape: the spheres formed resemble crystallization centers sized up to 100 μm. 

According to XRD, SEM, and IR spectroscopy, 100% DCPD powders were obtained with a unit cell volume (UCV) of 520 Å, consisting of large particles of 80–100 microns in size. Each powder particle was found to be an agglomerate of DCPD crystals at the micro level. Optimal conditions for the DCPD transformation process were experimentally substantiated: an increase in pH value above 5.5 ± 0.05 slows down the precipitation process or completely inhibits it, a decrease in pH less than 5.2 promotes the formation of the dicalcium phosphate anhydrous phase (DPA), which is toxic to the body. Reducing the processing time makes it impossible to obtain single-phase material.

DCPD coating on DBM matrices was obtained by precipitation of aqueous solutions of phosphate salts and sodium acetate. DBM was immersed in a solution at physiological temperature, but with different concentrations of calcium ion source, calcium chloride, from 1 to 2 M. 

The addition of 1 M calcium chloride concentration solution resulted in an uneven growth of DCPD crystals over the surface (Figure 4b,e), which does not fulfill the requirements for a composite biomaterial, since large crystals are detrimental for biointegration and adhesion, but can contribute to a more uniform distribution of the coating over the entire matrix volume and opportunities for further chemical transformation of DCPD into OCP. Increasing the concentration of calcium chloride exponentially leads to a more intense growth of DCPD crystals and their surface. Obtaining DCPD, at 25 °C, at a concentration of 2 M calcium chloride, allows for a uniform coating of the matrix with a layer of calcium phosphate. A further increase in the calcium chloride concentration, in the system under study, is inexpedient because the rate of the coating deposition process increases significantly, which leads to layer deformation, poor adhesion, and attachment to the DBM surface.

The IR spectroscopy data of the DCPD coatings obtained by this method correlate with previously published literature data [43,52,53] (Figure 5, Table 1). The IR spectrum of DCPD shows characteristic bands that occur around 3541, 3480, 3283, and 3166 cm^−1^ present a O–H stretching of lattice water molecules. The band at around 2930 cm^−1^ belongs to a (P)O–H stretching vibration mode. At 1649 cm^−1^ is the vibration of H–O–H bending of lattice molecules, mode at 1219 cm^−1^ presents a P–O–H in-plane bending mode. Bands at 1135, 1059, and 987 cm^−1^ present a P–O stretching mode. Vibration mode at 876 cm^−1^ presents a P–O(H) stretching mode. Vibration mode at 784 cm^−1^ presents a P–O–H out-of-plane bending mode. The water vibrations mode is at 662 cm^−1^. Bands that occur at 576 and 525 cm^−1^ belong to the O–P–O(H) bending modes. The obtained data for brushite are in good agreement with the results obtained by XRD. 

Figure 5a depicts the XRD patterns. The mineralogy of DCPD confirms that this precipitate, produced after mixing solutions with a Ca:P molar fraction 1:1, is pure brushite [31,32,33,34]. It is observed that the crystals of brushite grow in proportion to the major planes, namely (021), (041), and (020). The XRD pattern of the sample denotes brushite’s monoclinic structure [12,35]. The crystal growth takes place primarily along the (020) crystallographic plane [4], as evidenced by the peak at 2-theta 11.7°.

### 3.3. In Vitro Biocompatibility Evaluation Results

#### 3.3.1. Comparative Analysis of DBM Cytotoxicity

Cytotoxic properties of the three-dimensional DBM matrix, as well as cell adhesion to its surface, were determined by confocal microscopy, followed by digital processing of the obtained images. Figure 6 shows a micrograph of MG-63 cells cultured on a coverslip (control) and on DBM samples, 24 h (Figure 6a–c) and 72 h (Figure 6d–f) after seeding. The number of dead cells was found to be 6.3 ± 3.7% under control conditions after 24 h (Figure 6a). Twenty-four hours after cell seeding on DBM, all cells were attached to the DBM surface (Figure 6b,c) and the number of dead cells was 8.4 ± 7.1%, and significantly did not differ from control conditions.

After 72 h of cultivation under control conditions, the cells reached confluently, and the number of dead cells was 10.4 ± 1.5% (Figure 6d). Analysis of DBM samples after 72 h (Figure 6d–f) showed the smooth settlement of cells on the matrix surface. The number of dead cells was 11.7 ± 4.1%, which was also not significantly different from the control. All of the above suggests the absence of the cytotoxic effect of DBM, as well as free cell adhesion on its surface.

#### 3.3.2. Comparative Analysis of Cytotoxicity of Calcium Phosphate Compounds

The cytotoxicity of calcium phosphate compounds selected for remineralization was studied in the human osteoblast-like line MG-63 model, based on cell staining with Hoechst 33342 fluorescent dye (stains nuclei of living and dead cells blue), propidium iodide (stains nuclei of dead cells red) and calcein AM (stains cytoplasm of living cells green). DCPD obtained by low-temperature synthesis, and standard hydroxyapatite obtained by high-temperature synthesis, were used as comparison samples. Cells seeded on a culture plate were used as a control.

When CP was added to the human osteoblast-like line MG-63 for 24 h and 72 h, the cytotoxicity of DCPD did not differ from the control values (Figure 7a,b). Sintered HAp, both after 24 h (Figure 4a) and after 72 h (Figure 7b), exhibited a cytotoxic effect, starting from the concentration of 3 mg/mL, and this toxic effect increased up to 36.6 ± 4.9% cell death at a concentration of 10 mg/mL (Figure 7a,b).

Thus, based on the data obtained, a low-temperature DCPD obtained in 0.5 M NH_4_H_2_PO_4_ solution with 0.9 M CaCl_2_, without stirring for 24 h, at 35 ± 2 °C, was selected for further remineralization of the obtained DBM.

### 3.4. Results of Efficiency Evaluation of Three-Dimensional DBM Remineralization

#### Morphology Assessment of Remineralized Bone Matrices and DCPD Powder

SEM of DCPD powder samples established that the powder particles are lamellar crystals, between 20 and 100 μm long. (Figure 4b,e). DCPD is characterized by a lamellar particle morphology, with plate widths ranging from 10 to 35 μm and thicknesses varying up to several micrometers, depending on the particle size.

SEM of remineralized DBM samples revealed a uniform distribution of calcium deposits on the surface of DBM collagenous trabeculae, forming rounded hemispherical formations of the “rose flower” type characteristic of DCPD (Figure 4a,c–f). Precipitation from solutions on the polymeric DBM matrix leads to a reduction in the size of DCPD particles, which may be due to heterogeneous nucleation on collagen trabeculae [54]. Increasing solution concentration or temperature during deposition results in multicomponent systems with a heterogeneous microstructure. Lowering the temperature reduces the rate of processes, the transformation does not proceed completely, and the formation of multicomponent phase composition of powders is observed. A similar pattern was found in the deposition of HAp in gelatin solution, where the particle size decreased with increasing polymer concentration [55].

Histological analysis of remineralized DBM showed complete precipitation of calcium deposits directly on the trabecular surface, without their penetration into the main thickness of the collagen matrix, i.e., absence of intertrabecular mineralization and superficial precipitation of calcium deposits (Figure 4c, inset).

### 3.5. Results of the Assessment of Biocompatibility, Osteoinductive, and Osteoinductive Potential of Remineralized Three-Dimensional DBM in a Model of Ectopic Implantation in Rats

#### 3.5.1. Results of DCPD Implantation

After seven weeks of implantation, pure DCPD samples showed a complete resorption of the implanted materials, without any signs of encapsulation, aseptic calcinosis of healthy tissues, fibrosis, or formation of local adhesions in the peri-implant bed, i.e., the place of material implantation remained unchanged from the surrounding healthy tissue of rat recipients both at macro- and microscopic levels. Due to the absence of both positive and negative effects, the results of the histological analysis are not presented. The data obtained indicate a pronounced biocompatibility and a fairly rapid rate of resorption of low-temperature DCPD in experimental animals, expressed in the complete recovery of implanted bed tissues within seven weeks after implantation of the materials.

#### 3.5.2. Implantation Results of Pure DBM Samples

The high biocompatibility of the implanted bone-collagen matrices and the complete absence of osteoinductive and osteogenic potential of these materials were revealed after seven weeks of implantation (pure DCM without inspiration). Preservation of the collagen matrix ultrastructure after decellularization and demineralization of bone tissue was manifested by the absence of both enzymatic and cell-mediated resorption of implanted DBM. The high biocompatibility of DBM was manifested by the complete absence of signs of leukocytic invasion into the implanted material, the absence of giant foreign body cells (GFBC) and degenerated mast cells in the periphery of the samples, extremely weak sprouting of reactive altered connective tissue inside the sample from per implant bed tissue, and complete involution of the fibrous capsule by the studied period (Figure 8a,b). There were also no indications of DBM mineralization, both physiological (intertrabecular) biomineralization and pathological (utilization) calcinosis (Figure 8c). Signs of neocollagenesis were extremely weak and observed only at the periphery of the sample, and were initiated by peri-implant bed remodeling processes (host tissue influence), but not by DBM itself.

Thus, it has been established that both low-temperature DCPD and purified three-dimensional collagen DBM do not have osteoinductive or osteogenic potential themselves, and can act as an osteoconductor at most, without preventing the attachment, migration, and proliferation of recipient cells. 

#### 3.5.3. Subsubsection Results of Implantation of DBM Samples Remineralized by DCPD Layer 

In contrast to the DBM and DCPD groups separately, completely different/opposite results were obtained for remineralized DBM. The remineralized samples showed pronounced processes of haptotaxis, stimulation of proliferation, and synthetic activity of cells inside the implanted materials (Figure 8d), accompanied by pronounced processes of DBM remodeling (Figure 8d, arrows), as well as de novo structured neocollagenesis with the construction of new trabecula-like structures between trabecula ossea and lamellae of initial DBM (Figure 8e), indicating a pronounced synthetic activity of osteoblasts differentiated in the matrix. No histio-lymphocytic cells, mast cells, or GFBC were observed among the migrated cells; all the cells observed in the matrix belonged to mesenchymal cells. At the same time, no adipocytic cell differentiation was observed. For remineralized materials, there was a significant increase in the degree of calcification of the DBM matrix, while the nature of biomineralization was purely physiological (intratrabecular), with the active invasion of calcium deposits into the trabecular thickness and without any signs of destruction and defragmentation of the mineralized matrix by osteoclast-like cells; thus, no signs of utilization calcinosis were observed (Figure 8f). Bioimaging of the obtained histological images revealed that the degree of intratrabecular DBM mineralization increased by 3.37 times (cf. Figure 4c and Figure 8f). In addition to the above, the newly formed collagen matrix contained newly formed definitive blood vessels, which also indicates that immature osteoblasts, differentiated on the matrix, produce not only components of the bone matrix, but also factors of angiogenesis [56,57].

In Figure 9, under higher enlargement, it is seen that the construction of new trabeculae is carried out directly by the recipient cells (Figure 9a,b, arrows), with intrafibrillar biomineralization carried out centripetally (from the outside deep into the trabeculae).

## 4. Discussion

The development of biomimetic bone grafts, equal in effectiveness to bone autografts, is one of the most demanded tasks of modern tissue engineering [58]. The era of high-temperature, no-resorbable calcium phosphate ceramics is over due to the accumulation of sufficient clinical data on the lack of the necessary regenerative effect with the triggering of the typical reaction to a foreign object in the recipient’s body [59,60,61,62]. Currently, in the global scientific community, there has been a paradigm shift in the development of osteoplastic materials based on CPC: instead of creating load-bearing bioinert structures, the main task has become to influence the very process of material-associated reparative osteogenesis, there is a transition to stimulation of regenerative processes on the implanted material by involvement and differentiation of precursor cells into the osteoblast phenotype and maintaining the osteoclasts/osteoblasts balance. Perfect regenerative material should not pass the stage of complete (utilization) resorption, but should be remodeled in the recipient’s body and incorporated into the bone with the formation (restructuring) of personified bone tissue. This approach to the creation of the material allows to increase efficiency and also reduce the period of bone tissue regeneration. 

Various approaches have been proposed to increase the biological osteostimulating effect of bone implants, such as chemical modification of CP (substitution by cations, change of synthesis conditions etc.), and combination of scaffolds with bioactive molecules, such as growth factors, cytokines, peptides or small molecules targeting bone precursor cells, bone formation and metabolism or use cell-based strategies with progenitor cells combined or not with active molecules [63,64,65,66,67,68,69,70,71,72,73,74,75,76]. However, most of the proposed methods are excessively labor-intensive or expensive to produce, which reduces the availability of such materials to their main consumer, the average patient. The use of self-regulated natural DBM and the transition to the synthesis of calcium phosphate compounds under physiological conditions using bioactive bone hydroxyapatite precursors, remineralized on this DBM, open up new perspectives for the development and application of osteoplastic materials with high regenerative potential. Based on the experimental data obtained, it is assumed that the maximum preservation of the ultrastructure and microarchitectonics of the collagen bone matrix is a necessary condition for the physiological deposition of CPC on bone collagen, and provides a combined synergistic osteostimulating effect. Collagen fibrils are known to consist of an ordered array of rod-like molecules, which are packed in parallel and staggered lengthwise, at 67 nm in regard to each other, creating alternating areas of high and low packing density along the fibril axis direction, called Overlap Zone (~27 nm) and Hole Zone (~40 nm), respectively. Directly in the Hole Zones, amorphous calcium phosphates are precipitated with their further transformation into hydroxyapatite [77]. This process occurs “automatically” under the condition of supersaturation of Ca^2+^ and Ps created by the mineralizing osteoblasts with the help of microvesicles. All of the above indicates that the collagen fibril, due to its unique structure and spatial packing, directly controls important mineralization processes, such as providing calcium and phosphate delivery routes, as well as providing areas for the early nucleation of HAp crystals, determining the spatial distribution and topology of mineral binding [56]. 

The precursors of hydroxyapatite (HAp) formation, dicalcium phosphate dihydrate and octacalcium phosphate (OCP), are promising calcium phosphates for the biomineralization study. What they have in common is biocompatibility, that is, the absence of a negative reaction of the body to the implanted material and chemical stability concerning the biological medium. They act as centers of crystallization of hydroxyapatite of bone tissue and dental enamel, under conditions of deposition from the internal medium of the body, resulting in the formation of precipitated amorphous HAp, and subsequently HAp with a high degree of crystallinity. With low-temperature approaches, HAp formation is much slower than OCP, or DCPD, and while the simultaneous phase formation occurs during deposition, most of the kinetically favorable phase can be produced on the material, even though it has a much lower thermodynamic driving force. The advantage of the method of calcium phosphate deposition from solutions is the usage of physiological temperatures, which allows the incorporation of biological agents (proteins, growth factors) into the material, while keeping their biological activity. Additionally, using such an application approach, the physicochemical and morphological properties of the coatings do not depend on the surface geometry, or the three-dimensional structure of the material being modified. Research and application of such materials will make it possible to achieve quality osteogenesis when replacing critical defects of complex geometry, and reduce the incidence of complications.

## 5. Conclusions

The research described in this article involves the synthesis of new composite biomimetic remineralized three-dimensional collagen materials, with a significant osteostimulating effect based on natural high-purity bone-collagen de-mineralized bone matrices and DCPD with low production costs. Coating of DPCD (remineralization) on natural high-purity bone-collagen demineralized bone matrices, with preserved fibrillar structure as a biocompatible porous bioactive matrix by the low-temperature method in water solutions, leads to the formation of DCPD with monocrystalline lattice, with a uniform surface distribution and throughout the volume of the porous sample.

Evaluation of biocompatibility, biointegration, and bioactivity of the obtained materials in vitro and in vivo confirms the excellent biocompatibility of the matrix. the final material has no cytotoxic effect and promotes cell migration, adhesion and cell proliferation, with pronounced signs of physiological mineralization and matrix remodeling in the recipient body. 

## Figures and Tables

**Figure 1 biomimetics-08-00091-f001:**
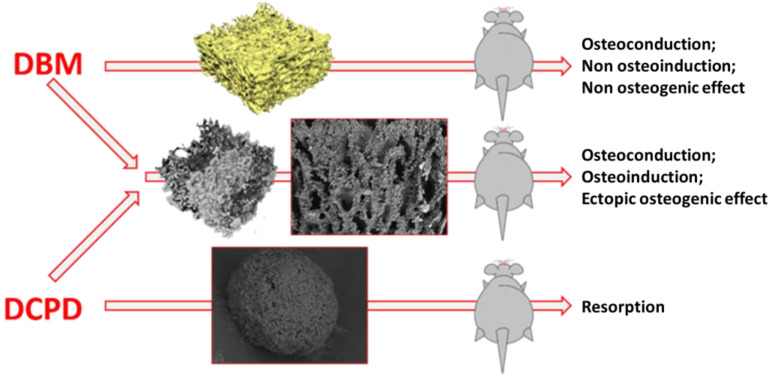
Scheme of the in vivo experiment. At the first stage ((**left part of the figure**), red text), highly purified DBM samples (**upper arrow**) and DPCD powder samples (**lower arrow**) were obtained, each which was separately implanted under the skin (ectopically) in rats. Simultaneously, DBM samples, remineralized by DPCD deposition technology, were obtained directly on the surface of the DBM bone trabeculae throughout its porous three-dimensional structure (**middle arrow**), with subsequent implantation of these DBM samples, remineralized by DPCD, also under the skin in rats. As can be seen from the diagram above, pure DBM samples showed osteoconductive, but not osteoinductive and osteogenic properties. Pure DPCD samples resorbed rather quickly in the animal body, while DBM samples remineralized by DPCD deposition showed pronounced osteoconductive and osteoinductive properties, as well as signs of full osteogenic potential.

**Figure 2 biomimetics-08-00091-f002:**
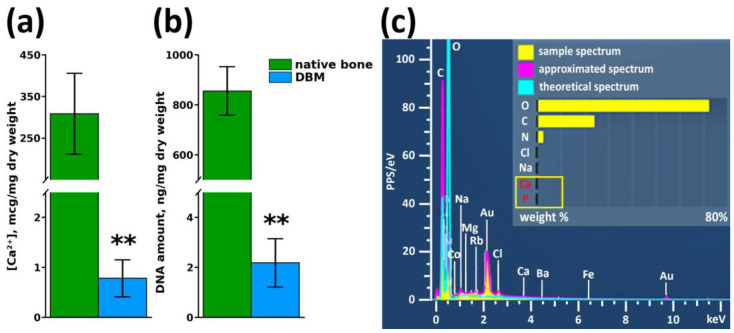
Results of DBM purity assessment after demineralization and decellularization procedures. (**a**) Calcium content in samples before and after demineralization, ** *p* < 0.01; (**b**) Residual cell DNA content in DBM before and after decellularization, ** *p* < 0.01; (**c**) Elemental surface analysis of DBM samples after demineralization and decellularization procedures.

**Figure 3 biomimetics-08-00091-f003:**
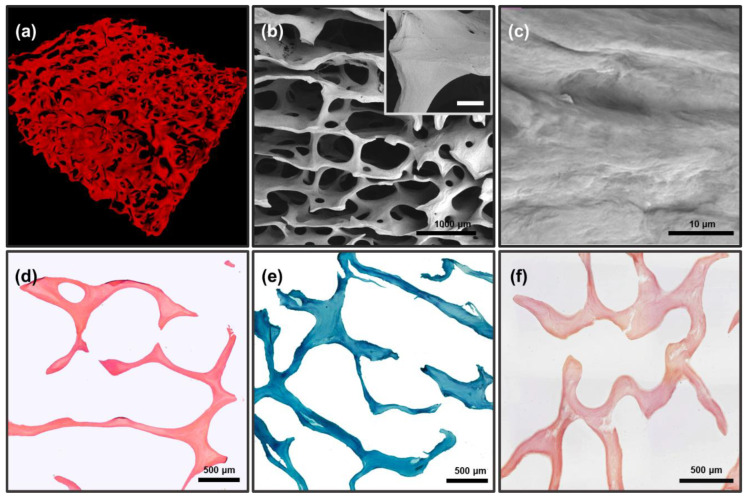
Results of morphological analysis of DBM after demineralization and before remineralization; (**a**) microCT image of DBM; (**b**,**c**) SEM images of DBM surface: (**b**) general view of the three-dimensional matrix, internal insets: an increased fragment of a trabecula ossea, scale 100 μm; (**c**) enlarged fragment of the collagenous trabecular surface. (**d**,**e**) Differential histochemical analysis of DBM samples; light microscopy: (**d**) hematoxylin and eosin staining (H&E), absence of cell nuclei (colored blue), erythrocytes (colored red), and muscle tissue (colored pink); (**e**) Lilly’s trichrome staining (collagen colored blue, absence of muscle and other tissues colored red and cell nuclei colored brown); (**f**) alizarin red S and toluidine blue staining (McGee-Russell method in Dahl’s modification, absence of calcium deposits colored orange-red, cell nuclei colored dark blue and background colored light blue).

**Figure 4 biomimetics-08-00091-f004:**
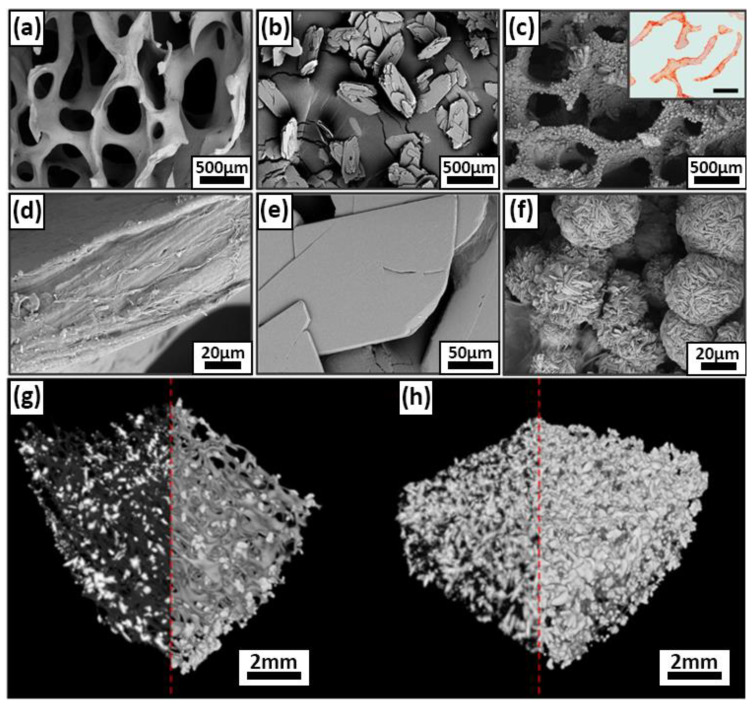
Results of morphological analysis of DPCD and remineralized DBM samples; (**a**,**d**) SEM of DBM surface before DPCD remineralization; (**b**,**e**) SEM of DPCD: (**c**,**f**) SEM of remineralized DBM, internal insets, a cross-section of a remineralized DBM, alizarin red S staining (McGee-Russell method, calcium deposits are colored orange-red, scale 500 μm); (**g**,**h**) micro CT three-dimensional structure of DCPD coating on DBM at 1 M CaCl_2_ leads (**g**) and at 2M CaCl_2_ leads (**h**).

**Figure 5 biomimetics-08-00091-f005:**
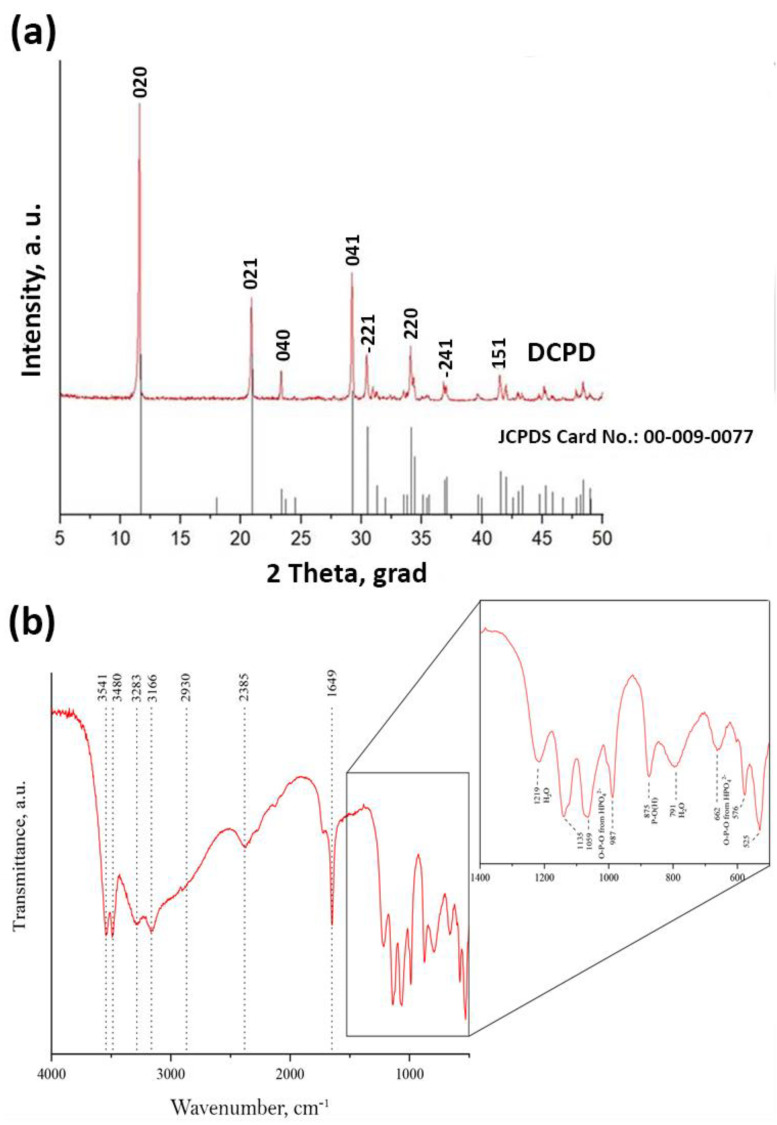
XRF data (**a**) and IR spectroscopy data (**b**) of DCPD coatings obtained by adding calcium chloride at a concentration of 2 M.

**Figure 6 biomimetics-08-00091-f006:**
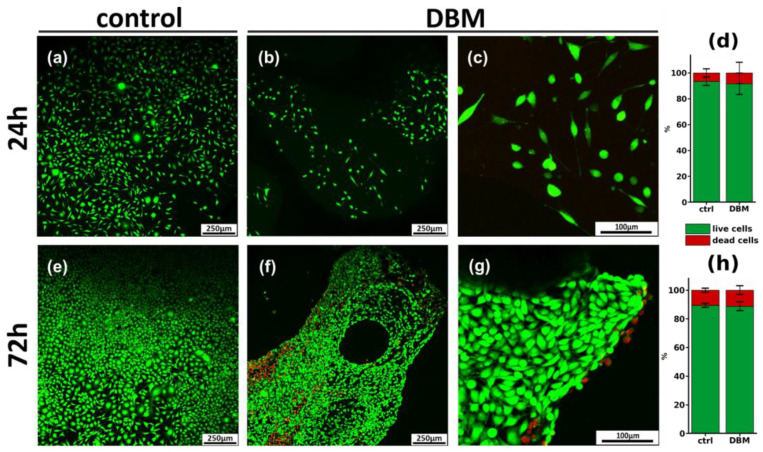
Results of cytotoxicity assays and morphological analysis of DBM samples. (**a**–**c**,**e**–**g**) Confocal micrographs of MG-63 cells after 24 h (**a**–**c**) and 72 h (**e**–**g**) of cultivation on the surface of DBM; live cells are stained with calcein AM (green), and dead cells are stained with PI (red). (**d**,**h**) Cytotoxicity of DBM after 24 h (**d**) and 72 h (**h**) incubation with MG-63 cells.

**Figure 7 biomimetics-08-00091-f007:**
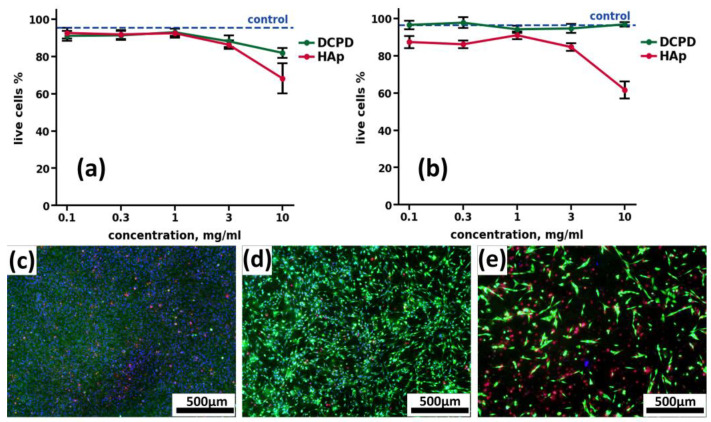
Results of cytotoxic effects of CPC cultured with human osteoblast-like line MG-63 cells at concentrations of 0.1, 0.3, 1.0, 3.0, and 10.0 mg/mL, after 24 h (**a**) and 72 h (**b**) of cultivation. Hoechst 33342, PI, calcein AM staining. ctrl—control without the addition of CPC; DCPD—dicalcium phosphate dihydrate; HAp—hydroxyapatite. (**c**–**e**) Fluorescence microimages of human osteoblast-like cells MG-63 in the control (**c**), in the presence of 10 mg/mL DCPD (**d**) and 10 mg/mL HAp (**e**) after 72 h of cultivation. Cell nuclei are stained with Hoechst 33342 (blue), the cytoplasm of live cells is stained with calcein AM (green), and nuclei of dead cells are stained with PI (red).

**Figure 8 biomimetics-08-00091-f008:**
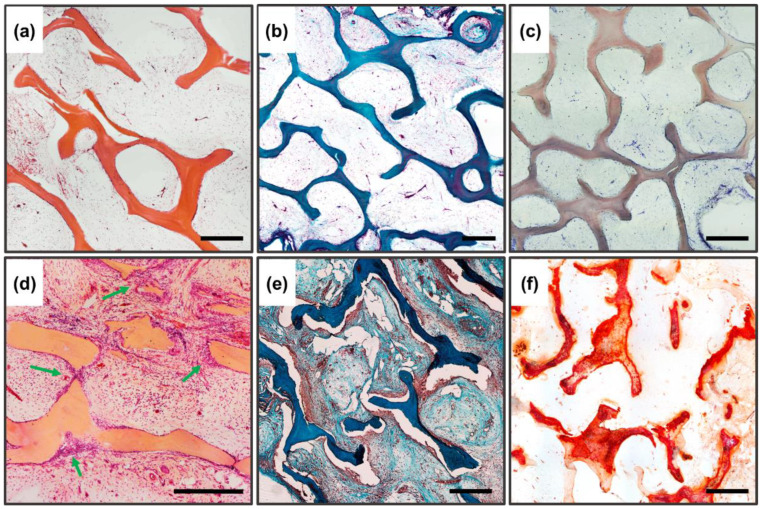
Histological analysis of implanted DBM samples of the control and experimental groups. (**a**–**c**) DBM without remineralization. (**d**–**f**) remineralized DBM. (**a**,**d**) Hematoxylin and eosin staining (H&E, cell nuclei colored blue, erythrocytes colored red, and muscle tissue colored pink); (**b**,**e**) Lilly’s trichrome staining (collagen colored blue, muscle and other tissues colored red, cell nuclei colored brown); (**c**,**f**) alizarin red S staining (calcium deposits colored orange-red), (**c**) due to the lack of staining with alizarin red S, a toluidine blue dyeing was performed to visualize the samples. Light microscopy; scale 500 µm.

**Figure 9 biomimetics-08-00091-f009:**
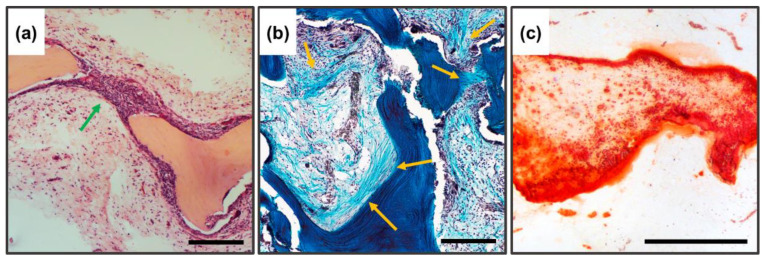
Histological analysis of implanted DBM samples of experimental groups, enlarged fragments. (**a**) Hematoxylin and eosin staining (H&E, cell nuclei colored blue, erythrocytes colored red, and muscle tissue colored pink); (**b**) Lilly’s trichrome staining (collagen colored blue, muscle and other tissues colored red, cell nuclei colored brown); (**c**) alizarin red S staining (calcium deposits are colored orange-red); the newly formed bone is indicated by the arrows. Scale 250 µm.

**Table 1 biomimetics-08-00091-t001:** Main Vibration Modes of Brushite (FTIR Spectroscopy).

Wavenumbers (cm^−1^)	Vibration Modes
3541–3480 3282–3166	O–H stretching of lattice water molecules
2930	(P)O–H stretching
2385, 1600–1720 (broad)	Combination H–O–H bending and rotation of residual free water
1649 (thin)	H–O–H bending of lattice water molecules
1219	P–O–H in-plane bending
1135, 1059, 987	P–O stretching
875	P–O(H) stretching
791	P–O–H out-of-plane bending
662	Water librations
576, 525	O–P–O(H) bending mode

## Data Availability

Data are available on request from the authors.

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
