# Peer review of "Biomimetic Remineralized Three-Dimensional Collagen Bone Matrices with an Enhanced Osteostimulating Effect"

_biomimetics, 2023, doi:10.3390/biomimetics8010091_

Round 1
Reviewer 1 Report
This paper presented a method to develop composite bioinspired bone grafts for reconstructive surgery by deposition (remineralization) on the surface of high-purity bone-collagen demineralized calcium phosphate layers. The assessment of the biocompatibility, biointegration and bioactivity of the obtained materials in vitro and in vivo demonstrated the promising biocompatibility of the matrix, providing evidence of its promotion towards cell migration, adhesion and proliferation, with noticeable signs of physiological mineralization and matrix remodeling in the receptors.
The current manuscript may be helpful for further related research after the following concerns are addressed:
1. Page 2, lines 70. This part of the discussion on the "Acellular Bone Extra-cellular matrix" is insufficient. It is only a brief list of its constituents and lacks the role that ECM plays in osteogenesis.
2. The introduction section lacks a brief description of the contents related to the proposed low-temperature DCPD remineralization in this paper. In addition, some innovative ideas should be highlighted.
3. Page 5, line 197. The caption of Figure 1 lacks the corresponding explanation of the figure. In addition, the interpretation of Figure 1 is lacking in the main text.
4. A few small mistakes. For example, Page 9, line 388. The quotation of Figure 5 in this part is not very relevant to the content of Figure 5.
5. Page 10, line 399. Please confirm the scale of Figure 4 (b).
6. Page 10, line 421. The details of Figure 5 are ambiguous and reconstruction is suggested.
7. Page 12, line 474. Why is the description of Figure 5, 6 and 7 preceded by Figure 4? The authors are encouraged to reorganize the order of the figures.
8. Page 13, line 522&523. The author mentioned that: "without preventing the attachment, migration, and proliferation of progenitor cells ", is there any evidence?
9. Page 15, line 581. "Various approaches have been proposed to increase the biological osteostimulating effect of bone implants" was mentioned, please enumerate these approaches briefly.
Author Response
Пожалуйста, посмотрите приложение.

Reviewer 2 Report
This manuscript aimed at creating an approach to develop bone grafts for reconstructive surgery, by deposition on the surface of high-purity bone-collagen demineralized calcium phosphate layers (DCPD). The author team utilized multiple characterization methods like IR, SEM, uCT, and X-ray to characterize the control (DBM) and coated DBM (DCPD-DBM); Cell culture and confocal were used to demonstrate the biocompatibility in vitro; Finally, a rat subcutaneous model was used to verify its remodeling and rebuilding performance.
Overall, the research strategy is clear and the methods used in this work are enriched and proper. However, one major concern about this manuscript arise from the reading of the draft:
1) Is the subcutaneous model proper to be used to verify the functionality of that bioinspired (DCPD-coated) DBM? After all, without exposing those grafts to the bone environment, it is not solid to guarantee their functionality or efficiency as the bone replacement matrix.
Besides this main point, other concerns/questions below arise from the reading of the paper. Please address them in the revised work for the publication of this manuscript in Biomimetic
· The reviewer has doubts that about the use of “Bioinspired” term. Your paper shows that DBM is more like “Bio-derived” instead of “bioinspired” since you didn’t synthesize the DBM by yourself. And how to get inspiration from the natural tissue to investigate the DCPD? Please clarify this used term.
· Line 145: What’s your control group? Please clarify it here
· Line 157: Is your CPC short for “Ca-P composite”? If so, please include the full name for this abbreviation. And also, if the DCPD is the only CPC using in your whole work, please use DCPD to replace all CPC in the method and results parts to avoid confusing the readers
· Section 2.3: What’s your 4th group? Also, it’s better to list a table to include the details of those 4 groups of samples.
· Line 242: Typo, 10^3
· Line 289: Why rat subcutaneous implant model was used? Why not use them under the bone environment? Any justification or references to demonstrate this model is valid for your purpose? Any gender information of selected rats disclosed in this section? Furthermore, this test is usually carried out at three months (12 weeks), why it was performed up to 7 weeks? Please, provide an explanation about the above questions.
· Line 295: How big the sample it is to be used to implant into rats?
· Figure 3, Line 357-359: 1) Please provide a zoom-in version of images to show the cell nuclei, erythrocytes and muscle tissue. It is recommended to use the arrow to indicate their locations.
· Line 345: Can you use the Micro CT to distinguish the DBM main structure with DCPD coating? In your whole work, the microCT was just used to acquire the 3d structure of DBM?
· Line 387: Why the 1M CaCl2 leads to uneven growth while 2M CaCl2 allows for a uniform coating? 2) Line 388, your figure 5 has XRD & IR images, it’s confusing to combine those two contents in one figure. Please separate them and also specify the figure to show the uneven coating.
· Figure 6: Cytotoxicity of DBM in figure d and h are similar, however, from Figure f vs b, g vs c, they behaved quite differently. Also, Figure f vs e also have a significant difference in the number of dead cells. Please clarify that and why does was Figure 6f also not significantly different from the control Figure 6e.
· Line 627: bioinspiration of CP or specifically, DCPD?
Author Response
Пожалуйста, посмотрите приложение.
